# Do Perceptually Aligned Gradients Imply Robustness?

## Abstract

Deep learning-based networks have achieved unprecedented success in numerous tasks, among which image classification. Despite these remarkable achievements, recent studies have demonstrated that such classification networks are easily fooled by small malicious perturbations, also known as adversarial examples. This security weakness led to extensive research aimed at obtaining robust models. Beyond the clear robustness benefits of such models, it was also observed that their gradients with respect to the input align with human perception. Several works have identified Perceptually Aligned Gradients (PAG) as a byproduct of robust training, but none have considered it as a standalone phenomenon nor studied its own implications. In this work, we focus on this trait and test whether *Perceptually Aligned Gradients imply Robustness*. To this end, we develop a novel objective to directly promote PAG in training classifiers and examine whether models with such gradients are more robust to adversarial attacks. We present both heuristic and principled ways for obtaining target PAGs, which our method aims to learn. Specifically, we harness recent findings in score-based generative modeling as a source for PAG. Extensive experiments on CIFAR-10 and STL validate that models trained with our method have improved robust performance, exposing the surprising bidirectional connection between PAG and robustness.

## 1 Introduction

AlexNet (Krizhevsky et al., 2012), one of the first Deep Neural Networks (DNNs), has significantly surpassed all the classic computer vision methods in the ImageNet (Deng et al., 2009) classification challenge (Russakovsky et al., 2015). Since then, the amount of interest and resources invested in the deep learning (DL) field has skyrocketed. Nowadays, such models attain superhuman performance in classification (He et al., 2016; Dosovitskiy et al., 2021). However, although neural networks are allegedly inspired by the human brain, unlike the human visual system, they are known to be highly sensitive to minor corruptions (Hosseini et al., 2017; Dodge & Karam, 2017; Geirhos et al., 2017; Temel et al., 2017; 2018; Temel & AlRegib, 2018) and small malicious perturbations, known as adversarial attacks (Szegedy et al., 2014; Athalye et al., 2018; Biggio et al., 2013; Carlini & Wagner, 2017b; Goodfellow et al., 2015; Kurakin et al., 2017; Nguyen et al., 2015). With the introduction of such models to real-world applications that affect human lives, these issues raise significant safety concerns, and therefore, they have drawn substantial research attention.

The bulk of the works in the field of robustness to adversarial attacks can be divided into two types – on the one hand, ones that propose robustification methods (Goodfellow et al., 2015; Madry et al., 2018; Zhang et al., 2019; Wang et al., 2020), and on the other hand, ones that construct stronger and more challenging adversarial attacks (Goodfellow et al., 2015; Madry et al., 2018; Carlini & Wagner, 2017a; Tramèr et al., 2020; Croce & Hein, 2020b). While there are numerous techniques for obtaining adversarially robust models (Lécuyer et al., 2019; Li et al., 2019; Cohen et al., 2019b; Salman et al., 2019), the most effective one is Adversarial Training (AT) (Madry et al., 2018). AT proposes a simple yet highly beneficial training scheme – train the network to classify adversarial examples correctly.

While exploring the properties of adversarially trained models, Tsipras et al. (2019) exposed a fascinating characteristic of these models that does not exist in standard ones – Perceptually Aligned Gradients (PAG). Generally, they discovered that such models are more aligned with human perception

than standard ones, in the sense that the loss gradients w.r.t. the input are meaningful and visually understood by humans. As a result, modifying an image to maximize a conditional probability of some class, estimated by a model with PAG, yields class-related semantic visual features, as can be seen in Fig. 1. This important discovery has led to a sequence of works that uncovered conditions in which PAG occurs. Aggarwal et al. (2020) revealed that PAG also exists in adversarially trained models with small threat models, while Kaur et al. (2019) observed PAG in robust models trained without adversarial training. While it has been established that robust models lead to perceptually aligned gradients, more research is required to better understand this intriguing property.

In this work, while aiming to shed some light on the PAG phenomenon, we pose the following reversed question – *Do Perceptually Aligned Gradients Imply Robustness?* This is an interesting question, as it tests the similarity between neural networks and human vision. Humans are capable of identifying the class-related semantic features and thus, can describe the modifications that need to be done to an image to change their predictions. That, in turn, makes the human visual system "robust", as it is not affected by changes unrelated to the semantic features. With this insight, we hypothesize that since similar capabilities exist in classifiers with perceptually aligned gradients, they would be inherently more robust.

To methodologically test this question, we need to train networks that obtain perceptually aligned gradients without inheriting robust characteristics from robust models. However, PAG is known to be a byproduct of robust training, and there are currently no ways to promote this property directly and in isolation. Thus, to explore our research question, we develop a novel PAG-inducing general objective that penalizes the input-gradients of the classifier without any form of robust training. However, this process requires access to "ground-truth" perceptually aligned gradients, which are challenging to obtain. We explore both heuristic and principled sources for such gradients. Our heuristic sources stem from the rationale that such gradients should point towards the target class. In addition, we provide in this work a second, principled approach towards creating such PAG vectors, relying on denoising score matching as used in generative models Song & Ermon (2019). We propose to estimate the gradient of the classification task for each input image as the difference between a conditional and unconditional score, both obtained by a pre-trained denoising network. This difference emerges from the Bayes rule, enabling theoretically justified distilled PAGs.

To validate our hypothesis, we first verify that our optimization goal indeed yields perceptually aligned gradients as well as sufficiently high accuracy on clean images, then evaluate the robustness of the obtained models and compare them to models trained using standard training ("vanilla"). Our experiments strongly suggest that models with PAG are inherently more robust than their vanilla counterparts, revealing that directly promoting such a trait can imply robustness to adversarial attacks. Surprisingly, not only does our method yield models with non-trivial robustness, but it also exhibits comparable robustness performance to adversarial training without training on perturbed images. These findings can potentially pave the way for standard training methods (*i.e.*, without performing adversarial training) for obtaining robust classifiers.

## 2 BACKGROUND

### 2.1 ADVERSARIAL EXAMPLES

We consider a deep learning-based classifier $f_\theta : \mathbb{R}^M \to \mathbb{R}^C$, where $M$ is the data dimension and $C$ is the number of classes. Adversarial examples are instances designed by an adversary in order to cause a false prediction by $f_\theta$ (Athalye et al., 2018; Biggio et al., 2013; Carlini & Wagner, 2017b; Goodfellow et al., 2015; Kurakin et al., 2017; Nguyen et al., 2015; Szegedy et al., 2014). In 2013, Szegedy et al. (2014) discovered the existence of such samples and showed that it is possible to cause misclassification of an image with an imperceptible perturbation, which is obtained by maximizing the network's prediction error. Such samples are crafted by applying modifications from a *threat model* $\Delta$ to real natural images. Hypothetically, the "ideal" threat model should include all the possible label-preserving perturbations, *i.e.*, all the modifications that can be done to an image that will not change a human observer's prediction. Unfortunately, it is impossible to rigorously define such $\Delta$, and thus, simple relaxations of it are used, the most common of which are the $\ell_2$ and the $\ell_\infty$ $\epsilon$-balls: $\Delta = \{\delta \ : \ \|\delta\|_{c \in \{2,\infty\}} \leq \epsilon\}$.

More formally, given an input sample $\mathbf{x}$, its ground-truth label $y$ and a threat model $\Delta$, a valid adversarial example $\hat{\mathbf{x}}$ satisfies the following: $\hat{\mathbf{x}} = \mathbf{x} + \delta$ $s.t.$ $\delta \in \Delta, y_{pred} \neq y$, where $y_{pred}$ is the prediction of the classifier on $\hat{\mathbf{x}}$. The procedure of obtaining such examples is referred to as an *adversarial attack*. Such attacks can be either untargeted or targeted. Untargeted attacks generate $\hat{\mathbf{x}}$ to minimize $p_\theta(y|\hat{\mathbf{x}})$, namely, cause a misclassification without a specific target class. In contrast, targeted attacks aim to craft $\hat{\mathbf{x}}$ in a way that maximizes $p_\theta(\hat{y}|\hat{\mathbf{x}})$ $s.t.$ $\hat{y} \neq y$, that is to say, fool the classifier to predict $\hat{\mathbf{x}}$ as a target class $\hat{y}$.

While there are various techniques for generating adversarial examples (Goodfellow et al., 2015; Carlini & Wagner, 2017a; Dong et al., 2018), we focus in this work on the Projected Gradient Descent (PGD) method (Madry et al., 2018). PGD is an iterative procedure for obtaining adversarial examples that operates as described in Alg. 1. The operation $Proj_\epsilon$ stands for a projection operator onto $\Delta$, and $\mathcal{L}(\cdot)$ is the classification loss, usually defined as the cross-entropy:

$$\mathcal{L}_{CE}(\mathbf{z}, y) = -\log \frac{\exp(\mathbf{z}_y)}{\sum_{i=1}^{C} \exp(\mathbf{z}_i)}, \tag{1}$$

where $\mathbf{z}_y$ and $\mathbf{z}_i$ are the classifier's logits for classes $y$ and $i$, respectively.

---

**Algorithm 1** Projected Gradient Descent

---

**Input**: classifier $f_\theta$, input $\mathbf{x}$, ground-truth class $y$, target class $\hat{y}$, threat model parameter $\epsilon$, step size $\alpha$, number of iterations $T$
$\delta_0 \leftarrow 0$
**for** *t from 0 to T* **do**
    **if** $\hat{y}$ *is not None* **then**
        $\delta_{t+1} = Proj_\epsilon(\delta_t - \alpha\nabla_\delta\mathcal{L}(f_\theta(\mathbf{x} + \delta_t), \hat{y}))$
    **else**
        $\delta_{t+1} = Proj_\epsilon(\delta_t + \alpha\nabla_\delta\mathcal{L}(f_\theta(\mathbf{x} + \delta_t), y))$
    **end**
**end**
$\mathbf{x}_{adv} = \mathbf{x} + \delta_T$
**Output**: $\mathbf{x}_{adv}$

---

## 2.2 ADVERSARIAL TRAINING

Adversarial training (AT) (Madry et al., 2018) is a learning procedure that aims to obtain adversarially robust classifiers. A classifier is considered adversarially robust if applying small adversarial perturbations to its input does not change its label prediction (Goodfellow et al., 2015). Such classifiers can be obtained by solving the following optimization problem:

$$\min_\theta \sum_{(\mathbf{x},y)\in D} \max_{\delta\in\Delta} \mathcal{L}(f_\theta(\mathbf{x} + \delta), y). \tag{2}$$

Intuitively, the above optimization trains the classifier to accurately predict the class labels of its hardest perturbed images allowed by the threat model $\Delta$. Ideally, $\mathcal{L}$ is the 0-1 loss, *i.e.*, $\mathcal{L}(\mathbf{z}, y) = \mathbf{I}(\texttt{argmax}_i(\mathbf{z}_i) = y)$ where $\mathbf{I}$ is the indicator function. Nevertheless, since the 0-1 loss is not differentiable, the cross-entropy loss, defined in Eq. (1), is used as a surrogate. In practice, solving this min-max optimization problem is challenging, and there are several ways to obtain an approximate solution. The most simple yet effective method is based on approximating the solution of the inner-maximization via adversarial attacks, such as PGD (Madry et al., 2018). According to this strategy, the above optimization is performed iteratively by first fixing the classifier's parameters $\theta$ and optimizing the perturbation $\delta$ for each example via PGD and then fixing $\delta$ and updating $\theta$. Repeating these steps results in a robust classifier. Since its introduction by Madry et al. (2018), various improvements to adversarial training were proposed (Andriushchenko & Flammarion, 2020; Huang et al., 2020; Pang et al., 2020; Qin et al., 2019; Xie et al., 2019; Zhang et al., 2019; Wang et al., 2020), yet in this work we will focus mainly on the basic AT scheme (Madry et al., 2018) for its simplicity and generality.

## 2.3 Perceptually Aligned Gradients

Perceptually aligned gradients (PAG) (Engstrom et al., 2019; Etmann et al., 2019; Ross & Doshi-Velez, 2018a; Tsipras et al., 2019) is a phenomenon according to which, classifier input-gradients are semantically aligned with human perception. This means, inter alia, that modifying an image to maximize a specific class probability should yield visual features that humans associate with the target class. Tsipras et al. (2019) discovered that PAG occurs in adversarially trained classifiers, but not in "vanilla" models. The prevailing hypothesis is that the existence of PAG only in adversarially robust classifiers and not in regular ones indicates that features learned by such models are more aligned with human vision. PAG is a qualitative trait, and currently, no quantitative metrics for assessing it exist. Moreover, there is an infinite number of equally good gradients aligned with human perception, *i.e.*, there are countless perceptually meaningful directions in which one can modify an image to look more like a certain target class. Thus, in this work, similar to (Tsipras et al., 2019), we gauge PAG qualitatively by examining the visual modifications done while maximizing the conditional probability of some class, estimated by the tested classifier. In other words, we examine the effects of a large-$\epsilon$ targeted adversarial attack and claim that a model has PAG if such a process yields class-related semantic modifications, as demonstrated in Fig. 1.

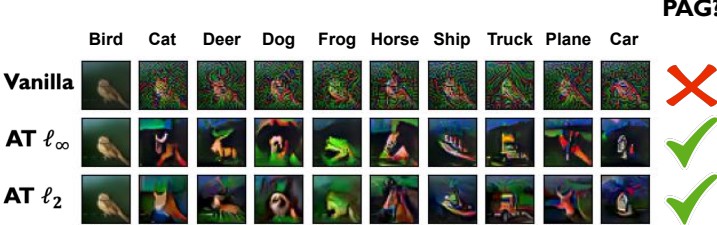

Figure 1: Visual demonstration of large-$\epsilon$ adversarial examples on "vanilla" and robust ResNet-18 trained on CIFAR-10 as a method to determine whether a model obtains PAG.

In recent years, PAG has drawn a lot of research attention which can be divided into two main types – an applicative study and a theoretical one. The applicative study aims to harness this phenomenon for various computer vision problems, such as image generation and translation (Santurkar et al., 2019), the improvement of state-of-the-art results in image generation (Ganz & Elad, 2021), and explainability (Elliott et al., 2021).

As for the theoretical study, several works aimed to better understand the conditions under which PAG occurs. Kaur et al. (2019) examined if PAG is an artifact of the adversarial training algorithm or a general property of robust classifiers. Additionally, it has been shown that PAG exists in adversarially robust models with a low max-perturbation bound (Aggarwal et al., 2020). To conclude, previous works discovered that training robust models leads to perceptually aligned gradients. In this work, we explore the opposite question – *Do perceptually aligned gradients imply robustness?*

## 3 Do Perceptually Aligned Gradients Imply Robustness?

As mentioned in Sec. 2.3, previous work has validated that robust training implies perceptually aligned gradients. More specifically, they observed that performing targeted PGD attacks on robust models yields visual modifications aligned with human perception. In contrast, in this work, we aim to delve into the opposite direction and test if training a classifier to have perceptually aligned gradients will improve its adversarial robustness.

To this end, we propose to encourage the input-gradients of a classifier $f_\theta$ to uphold PAG. Due to the nature of our research question, we need to isolate PAG from robust training and verify whether the former implies the latter. This raises a challenging question – PAG is known to be a byproduct of robust training. How can one develop a training procedure that encourages PAG without explicitly performing robust training of some sort? Note that a framework that attains PAG via robust training cannot answer our question, as that would involve circular reasoning.

We answer this question by proposing a novel training objective consisting of two elements: the classic cross-entropy loss on the model outputs and an auxiliary loss on the model's input-gradients.

We note that the input-gradients of the classifier, $\nabla_{\mathbf{x}} f_\theta(\mathbf{x})_y$, where $f_\theta(\mathbf{x})_y$ is the $y$-th entry of the vector $f_\theta(\mathbf{x})$, can be trained, since they are differentiable w.r.t. the classifier parameters $\theta$. Thus, given labeled images $(\mathbf{x}, y)$ from a dataset $D$, assuming we have access to ground-truth perceptually aligned gradients $g(\mathbf{x}, y_t)$, we could pose the following loss function:

$$\mathcal{L}_{total}(\mathbf{x}, y) = \mathcal{L}_{CE}\left(f_\theta(\mathbf{x}), y\right) + \lambda \sum_{y_t=1}^{C} \mathcal{L}_{cos}\left(\nabla_{\mathbf{x}} f_\theta(\mathbf{x})_{y_t}, g(\mathbf{x}, y_t)\right), \tag{3}$$

where $\mathcal{L}_{CE}$ is the cross-entropy loss defined in Eq. (1), $\lambda$ is a tunable regularization hyperparameter, $C$ is the number of classes in the dataset, and $\mathcal{L}_{cos}$ is the cosine similarity loss defined as follows:

$$\mathcal{L}_{cos}(\mathbf{v}, \mathbf{u}) = 1 - \frac{\mathbf{v}^\top \mathbf{u}}{\max(\|\mathbf{v}\|_2 \cdot \|\mathbf{u}\|_2, \varepsilon)}, \tag{4}$$

where $\varepsilon$ is a small positive value so as to avoid division by zero. Note that this loss considers the *direction* of the model's input-gradients without any requirement on their magnitude. This bodes well with the general goal of these gradients being *aligned* with human perception.

We emphasize that, in contrast to robust training methods such as (Madry et al., 2018; Cohen et al., 2019a), our scheme does not feed the model with any perturbed images and only trains on examples originating from the training set. Moreover, while other works (Ross & Doshi-Velez, 2018b; Jakubovitz & Giryes, 2018) suggest that penalizing the input-gradients' norm yields robustness, we do not utilize this fact since we encourage gradient alignment rather than having a small norm. Thus, our method is capable of promoting PAG without utilizing robust training.

After training a model to minimize the objective in Eq. (3), we aim to examine if promoting PAG in a classifier increases adversarial robustness. First, to verify that the resulting model indeed upholds PAG, we perform targeted PGD on test set images and qualitatively assess the validity of the resulting visual modifications. Afterwards, we test the adversarial robustness of the said model and compare it with vanilla baselines. If it demonstrates favorable robustness accuracy, we will have promoted an affirmative answer to the titular research question of this work.

However, one major obstacle remains in the way of training this objective: so far, we have assumed the existence of "ground-truth" model input-gradients, an assumption that does not hold in practice. While we hypothesize that these gradients should point in the general direction of the target class images, there is no clear way of obtaining point-wise realizations of them. In the following section, we begin by presenting practical and simple methods for obtaining approximations for these gradients, which we then use for training PAG-promoting classifiers. Next, we utilize score-based generative models for obtaining theoretically justified such gradients as a better source of PAG.

## 4 How are "Ground Truth" PAGs Obtained?

In order to train a classifier for minimizing the objective in Eq. (3), a "ground truth" perceptually aligned gradient $g(\mathbf{x}, y_t)$ needs to be provided for each training image $\mathbf{x} \in D$ and for each target class $y_t \in \{1, 2, \ldots, C\}$. Since a true such gradient is challenging to obtain, we instead explore few general pragmatic approaches for obtaining approximations for these PAGs, beginning with heuristic approaches then advancing to theoretically justified ones.

### 4.1 Target Class Representatives

As explained above, we aim to explore "ground truth" gradients that promote PAG without relying on robust models. To this end, we adopt the following simple premise: the gradient $g(\mathbf{x}, y_t)$ should point towards the general direction of images of the target class $y_t$. Therefore, given a representative of the target class, $\mathbf{r}_{y_t}$, we set the gradient to point away from the current image and towards the representative, *i.e.*, $g(\mathbf{x}, y_t) = \mathbf{r}_{y_t} - \mathbf{x}$. This general heuristic, visualized in Fig. 2, can be manifested in various ways, of which we consider the following:

**One Image (OI)**: Each representative should be chosen to reflect the visual features of its respective class. The simplest such choice that comes to mind is to choose $\mathbf{r}_{y_t}$ as an arbitrary training set image with label $y_t$, and use it as a global destination of $y_t$-targeted gradients. This *one image* approach satisfies the abstract requirements and provides simplicity, but it introduces a strong bias towards the arbitrarily chosen representative image, without considering the target class as a whole.

**Class Mean (CM)**: In order to reduce the bias towards a single image, we set $\mathbf{r}_{y_t}$ to be the mean of all the training images with label $y_t$. This mean can be multiplied by a constant in order to obtain an image-like norm. However, the *class mean* approach suffers from a clear limitation: a class' image distribution can be highly multimodal, possibly reducing its mean to a non-informative image.

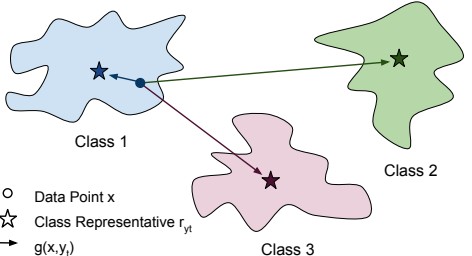

Class 1

Class 2

○ Data Point x
☆ Class Representative $\mathbf{r}_{y_t}$
→ g(x,$y_t$)

Class 3

**Nearest Neighbor (NN)**: As a possibly better trade-off, we examine a nearest neighbor approach – for each image $\mathbf{x}$ and each target class $y_t \in \{1, 2 \ldots, C\}$ we set the class representative $\mathbf{r}_{y_t}(\mathbf{x})$ (now dependent on the image) to be the image's NN amongst a limited set of samples from class $y_t$, using $L_2$ distance in the pixel space. More formally, we define

Figure 2: An illustration of the heuristic creation of perceptually meaningful gradients.

$$\mathbf{r}(\mathbf{x}, y_t) = \operatorname*{arg\,min}_{\hat{\mathbf{x}} \in D_{y_t} \text{ s.t. } \hat{\mathbf{x}} \neq \mathbf{x}} \|\hat{\mathbf{x}} - \mathbf{x}\|_2, \tag{5}$$

where $D_{y_t}$ is the set of sample images with class $y_t$. In practice, we sample $D_{y_t}$ to be a small number of *i.i.d.* training set images with class $y_t$.

## 4.2 SCORE-BASED GRADIENTS

Denoising diffusion probabilistic models (DDPMs) have recently emerged as an interesting generative technique (Sohl-Dickstein et al., 2015; Song & Ermon, 2019; Ho et al., 2020). Such models are capable of generating phorealistic images by performing an iterative process that starts from a Gaussian noise and follows the direction of the *score function*, defined as $\nabla_{\mathbf{x}_t} \log p(\mathbf{x}_t)$. Other works (Ho et al., 2022; Dhariwal & Nichol, 2021) have proposed to provide the class information to such networks, enabling them to model a class-condition score function $\nabla_{\mathbf{x}_t} \log p(\mathbf{x}_t|y)$. We provide additional details regarding DDPMs in Appendix B. we further examine these score functions and using the Bayes rule, we observe that the class-conditional score function can be factorized into

$$\nabla_{\mathbf{x}_t} \log p(\mathbf{x}_t|y) = \nabla_{\mathbf{x}_t} \log p(y|\mathbf{x}_t) + \nabla_{\mathbf{x}_t} \log p(\mathbf{x}_t), \tag{6}$$

leading to

$$\nabla_{\mathbf{x}_t} \log p(y|\mathbf{x}_t) = \nabla_{\mathbf{x}_t} \log p(\mathbf{x}_t|y) - \nabla_{\mathbf{x}_t} \log p(\mathbf{x}_t). \tag{7}$$

This equation brings forth a principled way to estimate the correct gradients for the expression $\log p(y|\mathbf{x}_t)$, which classification networks aim to output. By training a diffusion model that can optionally accept a class label (or act as unconditional), we obtain an estimation for "ground-truth" classifier input-gradients by a simple subtraction of conditional and unconditional outputs of the network. As can be seen in Eq. (7), the output is a subtraction of the unconditional score function from the class-conditioned one. To avoid the training of two separate diffusion models, we modify the model's architecture to account for both the conditional and unconditional cases. In particular, for a $C$-classes dataset, we train a single class-conditioned diffusion model with $C + 1$ classes, where the additional class represents the unconditional case. Instances of this class are drawn with probability $1/C$ and they originate uniformly from each of the $C$ classes. After training such a model, we use its outputs to obtain gradients according to Eq. (7).

## 5 EXPERIMENTAL RESULTS

In this section we empirically assess whether promoting PAG during classifier training improves its adversarial robustness at test time. We experiment using both synthetic and real datasets and present our findings in section 5.1 and sections 5.2 and 5.3, respectively.

## 5.1 A TOY DATASET

To illustrate and better understand the proposed approach and its effects, we experiment with a synthetic 2-dimensional dataset and compare our *nearest neighbor* method with the vanilla training scheme that minimizes the cross-entropy loss. We train a two-layer fully-connected classifier twice:

with our nearest neighbor method and without it. We then examine the obtained accuracies and visualize the decision boundaries. While both methods reach a perfect accuracy over the test set, the obtained decision boundaries differ substantially, as can be seen in Fig. 3. The baseline training method results in Fig. 3a yields dimpled manifolds as decision boundaries, as hypothesized by (Shamir et al., 2021) – the decision boundary of DNN is very close to the data manifold, exposing the model to malicious perturbations. In contrast, in Fig. 3b, the margin between the data samples and the decision boundary obtained using our approach is significantly larger than the baseline. This observation helps explain the following robustness result: our model achieves a $75.5\%$ accuracy on a simple adversarial PGD attack, whereas the baseline model collapses to $0.0\%$. The notion of "perceptually aligned" gradients admits a very clear meaning in the context of our 2-dimensional experiment – faithfulness to the known data manifold. Therefore, our empirical findings strongly attest that PAG imply robustness in the synthetic use case (see Appendix F.1 additional details).

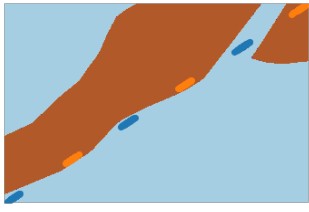
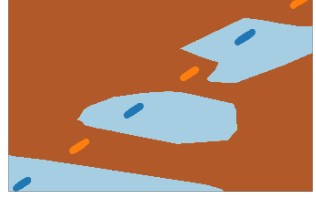

(a) Vanilla Training Scheme          (b) Our Training Scheme

Figure 3: Visualization of the decision boundary on a synthetic two-class dataset – the points are the test samples, and the background color represents the predicted class. Figures 3a and 3b present the decision boundary of a vanilla training method and ours, respectively.

## 5.2 CIFAR EXPERIMENTS

With the encouraging findings presented in Sec. 5.1, we now turn to conducting thorough experiments to verify if indeed promoting PAG can lead to improved adversarial robustness on a real dataset. We choose CIFAR-10 (Krizhevsky et al., 2014) as our main testing bed as it is the most common and studied dataset in the field of adversarial robustness. For methodologically answering our titular question, we train models to minimize Eq. (3) and follow a two-step evaluation procedure. First, we validate that models trained with our approach obtain PAG, and next, we test their adversarial robustness. As for examining if a model has PAG, we qualitatively probe whether modifying an image to maximize a certain class probability, estimated by a model, leads to a meaningful semantic change. For evaluating the robustness, we adopt AutoAttack (AA) (Croce & Hein, 2020a) under $\ell_\infty$ ($\epsilon = 8/255$) and the $\ell_2$ ($\epsilon = 0.5$) attacks.

In all the conducted experiments, we train a classifier using our proposed sources for ground-truth gradients – One Image (OI), Class Mean (CM), and Nearest Neighbor (NN) and the Score-Based Gradients (SBG). In addition, we train the same architectures using standard (Vanilla) and Adversarial Training (AT) (Madry et al., 2018) with $l_2$ ($\epsilon = 0.5$) and $l_\infty$ ($\epsilon = 8/255$) threat models. To control for the effect of the architecture used and better establish our empirical findings, we experiment with models from different architecture families – ResNet-18 (He et al., 2016) and ViT (Dosovitskiy et al., 2021). We provide additional implementation and experimental details in Appendix F.2. First, we show in Figure 4 that while vanilla models do not exhibit semantically meaningful changes, our approach does, as intended. Surprisingly, although our method is trained to have aligned gradients to some ground truth ones only on the data points, the model generalizes to have meaningful gradients beyond these points.

We proceed by quantitatively evaluating the performance on clean and adversarial versions of the test set, and show our CIFAR-10 results in Tab. 1. While the vanilla baseline is utterly vulnerable to adversarial examples, all the tested PAG-inducing techniques improve the adversarial robustness substantially while maintaining competitive clean accuracies. This strongly suggests that promoting PAG can improve the classifier's robustness in real image datasets. Moreover, as our method does not perform adversarial training, it is faster than AT by up to x6.14 (see Appendix G)

A closer inspection of the results indicates that our method performs better in the $L_2$ case over the $L_\infty$ one. We hypothesize that this stems from the Euclidean nature of the cosine similarity loss used to penalize the model gradients. Moreover, while both the heuristic-based and the theoretical-based

Table 1: Accuracy on the CIFAR-10 dataset using the ResNet-18 and ViT architectures.

| Method | Clean | | AutoAttack $L_2$ | | AutoAttack $L_\infty$ | |
|---|---|---|---|---|---|---|
| | RN-18 | ViT | RN-18 | ViT | RN-18 | ViT |
| Vanilla | **93.61**% | 80.51% | 00.00% | 00.87% | 00.00% | 00.01% |
| OI | 79.46% | 78.06% | 46.63% | 15.47% | 13.50% | 00.21% |
| CM | 81.41% | 78.98% | 47.25% | 13.73% | 11.24% | 00.17% |
| NN | 80.65% | 79.00% | 42.12% | 13.91% | 07.51% | 00.15% |
| SBG | 78.56% | **81.28**% | 55.39% | **57.80**% | 23.97% | 22.85% |
| AT $\ell_\infty$ | 82.49% | 62.20% | 56.57% | 42.80% | **37.59**% | **24.62**% |
| AT $\ell_2$ | 86.79% | 72.81% | **60.82**% | 42.99% | 19.63% | 08.13% |

"ground-truth" gradient sources substantially increase the adversarial robustness, the latter leads to significantly improved performance. It suggests that not only does PAG imply robustness, but there is also a positive correlation between the two – better sources for PAG lead to more robustness. To better study this effect, we use SBG with different values of $\lambda$ (*i.e.*, PAG regularization) and show that better PAG indeed leads to more robust models, as can be seen in Fig. 5. Interestingly, the robustification obtained by Score-Based Gradients is comparable to AT, without training on adversarial perturbations, potentially setting the foundations for non-adversarial methods for robust training. In addition, SBG outperforms AT $\ell_2$ on unseen attacks ($\ell_\infty$) on both RN-18 and ViT, without training on adversarial perturbed images.

Other fascinating results are the ones obtained on ViT, which is currently the best-performing vision model in computer vision. However, despite their tremendous popularity, ViTs are relatively unstudied in the field of adversarial robustness, which still mainly focuses on CNNs. Our findings show that off-the-shelf AT is significantly less effective on ViTs and decreases clean accuracy substantially. However, applying our method to ViT improves the robustness compared to AT while preserving the accuracy of natural images. Besides the improved quantitative performance, Figure 7 in Appendix E.2 shows that AT leads to inferior PAG compared to our method, which further attests to the strong bidirectional connection between PAG and adversarial robustness.

Finally, we report additional results on the CIFAR-100 dataset in Tab. 3 for proving the method's scalability and validating the connection between PAG and robustness in a more diverse real-world dataset.

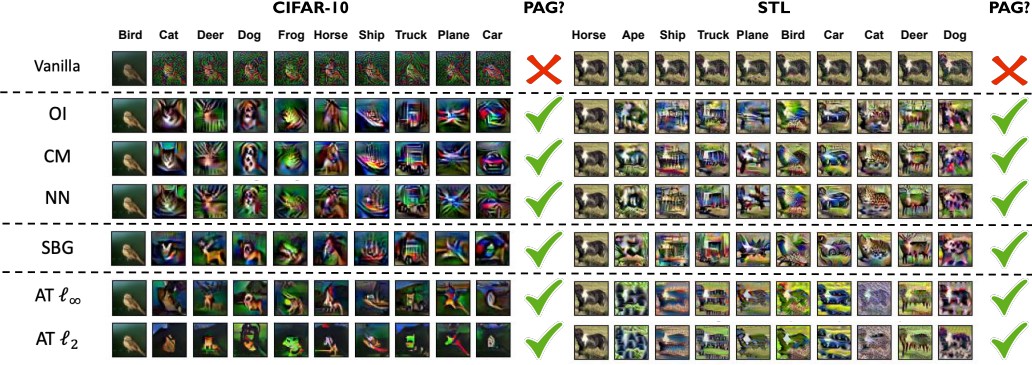

Figure 4: Perceptually Aligned Gradients phenomenon demonstrated by models trained with vanilla training (top), our method (middle), and AT (bottom), using ResNet-18 on CIFAR-10 and STL.

## 5.3 STL EXPERIMENTS

To better validate that our hypothesis holds in general, we test our approach on an additional dataset – STL (Coates et al., 2011), which contains images of a higher resolution of $96 \times 96$ pixels. Besides its resolution, we choose STL mainly due to its relatively small size – $5,000$ training and $8,000$ test images. While it is known that low data regimes are Adversarial Training's Achilles' heel, as it requires more training data (Schmidt et al., 2018; Zhai et al., 2019), we aim to investigate how our approach copes with such a challenging setup. We experiment with our different gradient sources using ResNet-18 and compare them to the standard and adversarial training baselines. As for the adversarial threat models, we use $\epsilon = 4/255$ for $L_\infty$ and $\epsilon = 0.5$ for $L_2$. We summarize our qualitative PAG results in Figure 4 and our quantitative ones in Tab. 2. As can be seen in the

results, while AT struggles to obtain decent results, both our heuristic and principled approaches significantly outperform it in clean and adversarial accuracy, while SBG is substantially better.

## 5.4 ENHANCING ADVERSARIAL TRAINING VIA PAG

Our empirical findings show that encouraging non-robust models to have PAG robustifies them. Moreover, we aim to check whether introducing our PAG-inducing objective into adversarial training (AT) can enhance its performance. Specifically, we apply it to AT $l_2$ trained on CIFAR-10 using ResNet-18 and find out that interestingly, it improves its $l_2$ and $l_\infty$ accuracy from $60.82\%$ and $19.63\%$, to $61.73\%$ and $23.52\%$, respectively. We provide additional details in Appendix F.3. These results indicate that our approach can operate both as a standalone robustification method and as an auxiliary loss for boosting the performance of existing robust optimization methods.

Table 2: Accuracy on STL using ResNet-18.

| Method | Clean | AA $L_2$ | AA $L_\infty$ |
|---|---|---|---|
| Vanilla | **82.60%** | 00.00% | 00.00% |
| OI | 71.29% | 57.91% | 29.65% |
| CM | 70.66% | 58.90% | 33.71% |
| NN | 70.16% | 60.02% | 36.21% |
| SBG | 74.79% | **65.96%** | **43.53%** |
| AT $\ell_\infty$ | 54.90% | 46.33% | 28.30% |
| AT $\ell_2$ | 54.99% | 46.04% | 23.33% |

Figure 5: Demonstration of the positive correlation between PAG and robustness. Higher $\lambda$ indicates more perceptually aligned gradients.

## 6 CONCLUSIONS AND FUTURE WORK

While previous work demonstrates that adversarially robust models uphold the PAG property, in this work, we investigate the reverse question – *Do Perceptually Aligned Gradients Imply Adversarial Robustness?* We believe that answering this question sheds additional light on the connection between robust models and PAG. To empirically show that inducing PAG improves classifier robustness, we develop a novel generic optimization loss for promoting PAG without relying

Table 3: Accuracy on CIFAR-100 using ResNet-18.

| Method | Clean | AA $L_2$ | AA $L_\infty$ |
|---|---|---|---|
| Vanilla | **74.36%** | 00.00% | 00.00% |
| CM | 58.89% | 19.94% | 02.78% |
| SBG | 55.94% | 29.25% | 08.24% |
| AT $\ell_\infty$ | 52.92% | 26.31% | **14.63%** |
| AT $\ell_2$ | 58.05% | **30.51%** | 08.03% |

on robust models or adversarial training and test several manifestations of it. Our findings suggest that all the manifestations of our PAG-inducing method improve the adversarial robustness compared to a vanilla model. Specifically, our Score-Based Gradients approach provides robustness on par with AT, which strongly suggests that PAG leads to robustness. Moreover, while AT requires a large amount of data, our approach outperforms it on low-data regimes. In addition, our score-based method shows better results than AT in ViT. We hope this work will lay the foundations for developing non-adversarial methods for obtaining robust models in the future.

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

# A    RELATED WORK

Recent works have explored properties of input gradients that improve adversarial robustness. The authors of Jakubovitz & Giryes (2018) demonstrate that regularizing the Frobenius norm of a classifier's Jacobian to be small improves robustness. Such a method is equivalent to regularizing the norm of each such gradient to be small, similar to Ross & Doshi-Velez (2018); Finlay & Oberman (2021). This line of work attests that requiring small gradient norms, regardless of their direction, leads to robustness. Moreover, none of these works promotes nor exhibits perceptually aligned gradients. On Contrary, we penalize over the direction of the gradients, regardless of their size – opposing to the above methods. By showing that our approach leads to PAG and improved robustness, our method can be viewed as an alternative input-gradient loss for improving robustness. Nevertheless, the main goal of our work is to better study PAG and its connection with adversarial robustness.

Another interesting work is ClusTR (Alfarra et al., 2020), which reveals an intrinsic connection between clustering and robustness. Moreover, the authors claim that clustering similar instances in the feature space encourages the network to learn semantically meaningful representations. Although related to our work, there are some main differences. First, as clustering is related to robustness, this approach cannot test our titular question. Moreover, ClusTR was not proven to have PAG or more semantical meaningful features, contrary to our work.

# B    DENOISING DIFFUSION PROBABILISTIC MODELS

Denoising diffusion probabilistic models (DDPMs) are a new fascinating generative approach (Sohl-Dickstein et al., 2015; Song & Ermon, 2019; Ho et al., 2020). Such methods achieve state-of-the-art results in image generation (Dhariwal & Nichol, 2021; Song et al., 2021; Vahdat et al., 2021), and were additionally deployed in several downstream tasks such as inverse problems (Kawar et al., 2021; 2022), image compression (Theis et al., 2022), image segmentation (Amit et al., 2021), image editing (Liu et al., 2021; Avrahami et al., 2022), text-to-image generation (Ramesh et al., 2022; Saharia et al., 2022), among others.

The core idea of these models, which are also known as score-based generative models, is to start from a random Gaussian noise image $\mathbf{x}_T$, and then iteratively denoise it into a photorealistic image $\mathbf{x}_0$ in a controlled manner. This process can also be interpreted as an annealed version of Langevin dynamics (Song & Ermon, 2019), where each iteration $t \in \{T, T-1, \ldots, 1, 0\}$ follows the direction of the *score function*, defined as $\nabla_{\mathbf{x}_t} \log p(\mathbf{x}_t)$, with an additional noise for stochasticity. Each intermediate image $\mathbf{x}_t$ can be considered a noisy version of a pristine image $\mathbf{x}_0$, with a pre-defined noise level $\sigma_t$. The score function can be estimated using a neural network trained for mean-squared-error denoising (Stein, 1981; Miyasawa, 1961; Efron, 2011). This estimation can also be generalized for denoising models conditioned on a class label $y$, obtaining $\nabla_{\mathbf{x}_t} \log p(\mathbf{x}_t|y)$ (Ho et al., 2022).

While diffusion models have been recently used in the context of adversarial robustness (Nie et al., 2022; Blau et al., 2022), these works were mostly focused on denoising an adversarially perturbed input to a vanilla classifier. In contrast, we propose a novel usage of diffusion model outputs as guidance for training classifier gradients. Our method, in turn, enjoys the added benefit of significantly faster runtime, as no iterative process is applied.

# C    ADDITIONAL ARCHITECTURES ABLATION

In this section, we provide the results of applying our method to additional architecture types. While we focus in Sec. 5 on skip-connection-based convolutional NN (ResNet-18) and an attention-based one (ViT), we turn to examine it on other types of architectures. Specifically, we apply it to VGG (Simonyan & Zisserman, 2014), a convolutional network without residual connection, and MLP Mixer (Tolstikhin et al., 2021), a top-performing dense architecture. We experiment with such architectures using the SBG approach on CIFAR-10 and report the results in Tab. 4.

For the MLP Mixer, we follow the CIFAR-10 adjusted implementation[1] and train it for 100 epochs using a batch size of 128. We apply SBG and set $\lambda = 0.5$. As for the VGG, we follow the

---

[1]https://github.com/omihub777/MLP-Mixer-CIFAR

implementation for CIFAR-10[2] and train for 100 epochs using a batch size of 64. Similarly to MLP Mixer, we use $\lambda = 0.5$. Moreover, we do so for both VGG-11, VGG-13 and VGG-16 to further study the depth effect. These ablation studies show that the connection between PAG and robustness is general and architecture-independent.

Table 4: Accuracy on CIFAR-10 using VGG and MLP Mixer

| Method | Arch. | Clean | AutoAttack $L_2$ |
|---|---|---|---|
| Vanilla | VGG-16 | **92.32**% | 00.20% |
| SBG | | 81.93% | **42.03**% |
| Vanilla | VGG-13 | **92.47**% | 00.11% |
| SBG | | 82.05% | **41.49**% |
| Vanilla | VGG-11 | **90.82**% | 02.50% |
| SBG | | 79.22% | **35.79**% |
| Vanilla | MLP-Mixer | **72.05**% | 00.50% |
| SBG | | 63.04% | **35.97**% |

## D  TINY IMAGENET EXPERIMENTS

To further verify if indeed PAG implies robustness, we apply our method to Tiny ImageNet, in addition to CIFAR-10, CIFAR-100, and STL. This dataset contains 100,000 images of size $64 \times 63 \times 3$ and 200 classes and is very diverse. In this section, we report the results of encouraging PAG using Class Mean on a ResNet-18 (we set $\lambda = 2$) and train for 50 epochs, and report the results in Tab. 5. These empirical findings testify to the connection between PAG and robustness in a more general content dataset and demonstrate our method's scalability.

Table 5: Accuracy on Tiny ImageNet using ResNet-18

| Method | Arch. | Clean | AutoAttack $L_2$ |
|---|---|---|---|
| Vanilla | ResNet-18 | **61.19**% | 02.37% |
| SBG | | 50.04% | **19.21**% |

## E  VISUALIZATION

### E.1  PAG GROUND-TRUTH VISUALIZATION

We provide visualization of our "ground-truth" sources for Perceptually Aligned Gradients in Figure 6. In the top three rows, we show the results of the heuristic-based methods. As these gradients derive from the subtraction of two images, a ghosting effect can be seen. However, in Score-Based Gradients, the modifications focus on the object, and features of the target class can be observed (i.e., horse features). The nature of SBG resembles the one in Adversarial Training, as can be seen in the bottom row – focus on the object itself.

### E.2  VIT ON CIFAR-10

We show a qualitative demonstration of the Perceptually Aligned Gradients results on CIFAR-10 using ViT in Figure 7. As can be seen, similar to ResNet's results, the vanilla model does not show PAG at all. The heuristic sources lead to some improvement, while our Score-Based Gradients approach leads to better PAG than AT. That demonstrates the connection between PAG and robustness, as SBG shows better PAG and improved adversarial robustness.

---

[2]https://github.com/chengyangfu/pytorch-vgg-cifar10

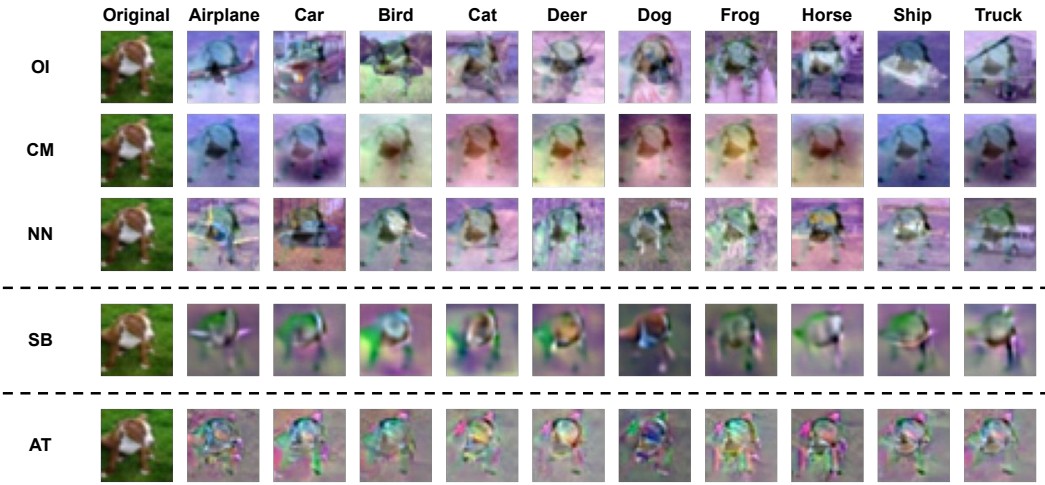

Figure 6: Visualization of PAG "ground-truth" gradients of different sources.

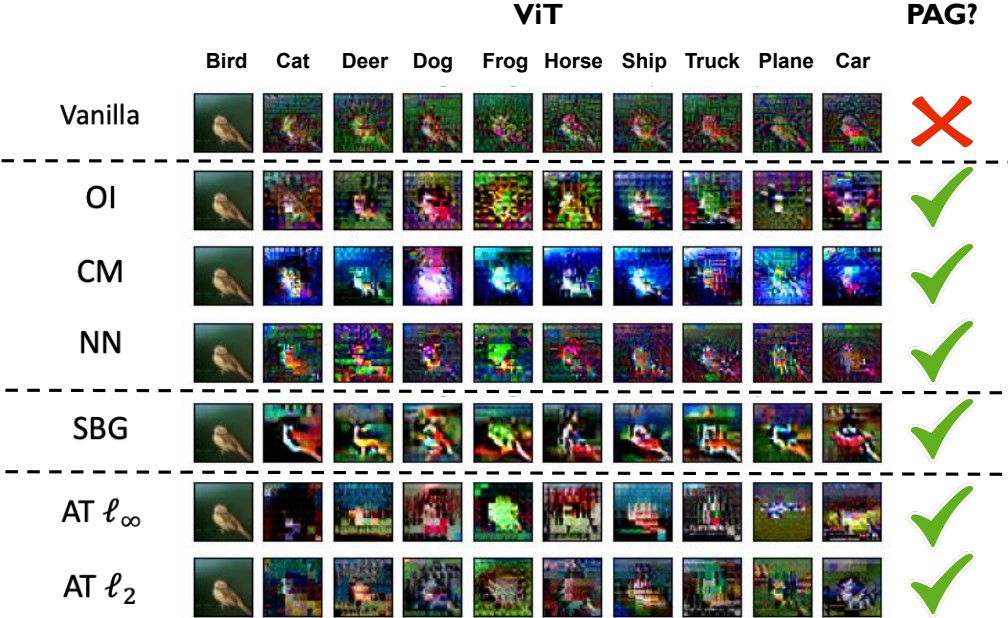

Figure 7: Visualization of PAG on CIFAR-10 dataset using ViT.

## F  IMPLEMENTATION DETAILS

### F.1  TOY DATASET

**Data**: We experiment with our approach on a 2-dimensional synthetic dataset to demonstrate its effects. To this end, we construct a dataset consisting of 6,000 samples from two classes, where each class contains exactly 3,000 examples. Our samples, $\mathbf{x} = [x_1, x_2]$, reside on the straight line $x_2 - 2x_1 = 0$ in the 2-dimensional space $\mathbb{R}^2$, where each class $y \in \{0, 1\}$ follows a Gaussian mixture distribution. Each class contains three modes, and each of them contains 1000 samples drawn from a Gaussian distribution ($x_1 \sim N(c, 1), x_2 = 2 * x_1$, where $c$ is the mode center). The modes centers are set to be $\{-50, -10, 30\}$ and $\{-30, 10, 50\}$. This way, the cardinal manifold assumption according to which high-dimensional images reside on a lower-dimensional manifold

Ruderman (1994) holds. We generate a balanced test set from the same distribution consisting of 600 samples and use it to evaluate performance.

**Architecture and Training**: We use a 2-layer fully-connected network $(2 \rightarrow 32 \rightarrow 2)$ with ReLU non-linearity. We train it twice – using the standard cross-entropy training and our proposed method with NN realization. We do so for 100 epochs with a batch size of 128, using Adam optimizer, a learning rate of 0.01, and the same seed for both training processes.

**Computational Resources**: We use a single Tesla V100 GPU.

**Evaluation**: As detailed in the paper, we test the performance of the models using standard and adversarial evaluation. For the standard one, we draw 600 test samples from the same distribution as the train set and measure the accuracy. As for the adversarial one, we use an $L_2$-based 10-step PGD with $\epsilon = 15$ and a step size of 2. Note that this choice of $\epsilon$ guarantees in our settings that the allowed threat model is too small for actually changing a sample of a certain class to the other one, making it a valid threat model.

### F.2 REAL DATASETS

**Data**: As for our real datasets experiments, we use CIFAR-10 and STL that contain images of size $32 \times 32 \times 3$ and $96 \times 96 \times 3$, respectively. For each realization, before the training procedure, we construct a dataset by computing $C$ targeted gradients for each training sample ($C = 10$ for CIFAR-10 and STL) for reproducibility and consistency purposes.

To obtain our Score-Based Gradients (SBG) we follow the implementation of (Nichol & Dhariwal, 2021)[3] for training a class-conditioned diffusion model for CIFAR-10 and STL datasets. We use their CIFAR-10 architecture for CIFAR-10, and their ImageNet architecture for STL, adapting the image size by a simple bicubic interpolation. In particular, for a $C$-classes dataset, we train a single class-conditioned diffusion model with $C + 1$ classes, where the additional class represents the absence of class information and thus models the unconditional score function. Instances of this class are drawn with probability $1/C$ and they originate uniformly from each of the $C$ classes. After training such a model, we use it to distill gradients according to Eq. (7).

**Training**: For both datasets, we train a ResNet-18 for 100 epochs, using SGD with a learning rate of 0.01, a momentum of 0.9, and a weight decay of 0.0001. In addition, we use the standard augmentations for these datasets – random cropping with padding of 4 and random horizontal flipping with a probability of 0.5. We use a batch size of 64 for CIFAR-10 and 32 for STL. As for the ViT, we use a an implementation adjusted to CIFAR-10[4], containing 6.3 million parameters. We present in Tab. 6 the best choices of $\lambda$ – the coefficient of our PAG promoting auxiliary loss term in all the tested datasets and methods. The values of $\lambda$ suggest that higher values should be applied for better gradient sources (*e.g.*, SBG's ideal $\lambda$ is higher than the heuristic methods).

As for our baselines, we use the same training hyperparameters mentioned above. Regarding the AT, we use 7 steps PGD using step size of $1.5 * \frac{\epsilon}{7}$, and follow the base implementation presented in (Zhang et al., 2019)[5] and extend it to $\ell_2$.

Table 6: Values of the hyperparameter $\lambda$.

| Method | CIFAR-10 | | STL |
| | RN-18 | ViT | RN-18 |
| --- | --- | --- | --- |
| One Image | 0.5 | 0.1 | 0.25 |
| Class Mean | 0.4 | 0.1 | 0.2 |
| Nearest Neighbor | 0.4 | 0.1 | 0.4 |
| SBG | 2 | 2 | 1 |

**Computational Resources**: We use two NVIDIA RTX A4000 16GB GPUs for each experiment.

---

[3]`https://github.com/openai/improved-diffusion`
[4]`https://github.com/omihub777/ViT-CIFAR`
[5]`https://github.com/yaodongyu/TRADES`

**Evaluation**: We use the de-facto standard evaluation library of AutoAttack (Croce & Hein, 2020a)[6].

**Code**: We attach our anonymized code to the submission, and will publicly release it along with our trained models upon acceptance.

### F.3  ENHANCING ADVERSARIAL TRAINING VIA PAG

To test whether our approach is capable of further enhancing adversarial training, we apply it as an auxiliary loss. We use ResNet-18, train it on CIFAR-10 using AT with 7-steps PGD and introduce our loss with $\lambda = 0.2$. We conclude the results in Tab. 7.

Table 7: Enhancing Adversarial Training via PAG

| Method | Clean | AutoAttack $L_2$ | AutoAttack $L_\infty$ |
|---|---|---|---|
| AT | **86.79** | 60.82 | 19.63 |
| AT + PAG | 85.54 | **61.73** | **23.52** |

## G  RUNTIME COMPARISON WITH ADVERSARIAL TRAINING

As our method does not compute adversarial example (an iterative process), it is faster than adversarial training. To quantify this, we conduct a runtime comparison using ResNet-18 and CIFAR-10 dataset and reveal that our method is faster than AT-PGD-7 and AT-PGD-20 by x2.13 and x6.14, respectively.

---

[6]https://github.com/fra31/auto-attack

