# OpenReview forum: "Do Perceptually Aligned Gradients Imply Robustness?"
_ICLR.cc/2023/Conference — Submitted to ICLR 2023_

### Official Review · Reviewer_snWm · 2022-10-20

**Confidence:** 4
**Correctness:** 4
**Technical Novelty And Significance:** 3
**Empirical Novelty And Significance:** 3
**Recommendation:** 6

**Clarity, Quality, Novelty And Reproducibility:**

Clarity: This paper is written clearly.
Quality: The quality is good
Novelty: The problem and the approach are both novel.
Reproducibility: The authors have provided the code in the appendix.


**Strength And Weaknesses:**

Strengths:
1. This paper asks an interesting question: is perceptually-aligned gradients also a sufficient condition for adversarial robustness. While previous work has shown that it is necessary, none has looked into if the reverse also holds true. Trying to answer this question would help the community understand the importance of former (perceptually-aligned gradients) to the latter (adversarial robustness).
2. The method proposed in section 3 is novel and can be a principled approach for future work as to how to induce percetually-aligned gradients during training.
3. The experiments are solid and the conclusion drawn from the experiments that perceptually-aligned gradients do imply adversarial robustness is interesting. It could benefit future work in this direction to rethink the importance of perceptually-aligned gradients.


Weaknesses:
1. It would be nice to report the training time of the proposed approach compared to adversarial training. It seems that it would require less backward pass compared to full PGD with many steps as used in AT. Thus it would be nice to have a formal comparison of the statistics of time spent.
2. I don’t quite follow the part about score-based gradients. So are you using the gradients computed from equation 7 as the new target g? It seems that you need two diffusion models, one being class-conditional and one being unconditional to estimate the two parts in the right hand side of equation 7.
I feel the first two paragraphs of section 4.2 can be moved into background or related work. It seems hard to follow which is exactly your contribution. I would elaborate more on how you obtain the gradients in equation 7.


**Summary Of The Paper:**

While previous work has established that adversarial robust models lead to perceptually-aligned gradients, this paper tries the answer if the reverse condition also holds that if a model has perceptually-aligned gradients, is it robust to adversarial examples? The answer given by this paper is yes. The crux of the approach is a new training scheme which explicitly optimizes for the alignment between current input gradients and perceptually-aligned gradients. The authors proposed several approaximations to achieve this gradient alignment. The proposed method is evaluated on CIFAR-10 and STL.

**Summary Of The Review:**

The paper is clearly written. I like the question this paper is trying to answer. The proposed approach can be a principled way to directly induce perceptually-aligned gradients. For now i am leaning towards accept.

---

> ### Author Response · Authors · 2022-11-19
> **Authors' reply**
>
> We thank the author for the constructive feedback. We address below every raised point and revise our manuscript accordingly.
>
> * We thank the reviewer for this suggestion – we report the requested statistics in Appendix G. As the reviewer stated correctly, we find out that our method is x6.14 times faster than AT on CIFAR-10 which uses the standard PGD-20.
>
> * Thanks for your suggestion – we have updated the manuscript according to your advice (added a DDPMs background section in the Appendix and edited section 4.2). Regarding the SBG, intuitively you are correct - two diffusion models are required. Nevertheless, as was specified in Appendix B.2., given a dataset with C classes, we introduce class C+1 that represents the unconditional case. This implementation trick enables us to utilize a single diffusion model for both the conditional (using the first C classes) and the unconditional one (using "class" C+1).

---

> > ### Comment · Reviewer_snWm · 2022-11-24
> > **Question regarding PAG ground-truth gradients from diffusion models**
> >
> > Hi authors,
> >    I really appreciate your response. Thanks for re-organizing section 4.2. Yet I still have one another question regarding this part. Specifically in equation 7, diffusion models would expect $x_t$ to be a noisy version of clean image, but i assume here you are just computing the score function for clean image, i am not sure if the gradient computed on clean image is meaningful here. I think a distinction should be made here in order to create confusion. That's my major concern here.
> >
> >    I am also reading this paper https://arxiv.org/abs/2207.12598 and the deviration of their classifier-free guidance seems very similar although the goal is different. Still, $x_t$ is supposed to be noisy images in diffusion models.

---

> > > ### Author Response · Authors · 2022-11-24
> > > **Authors answer**
> > >
> > > Dear reviewer snWm,
> > >
> > > You are absolutely correct -- in order to obtain meaningful score functions using a diffusion model, the input images need to be noisy. Thus, we indeed use noisy versions of the clean images in our experiments, denoted as $x_t$ in equation 7 and further explained in Appendix B. However, we will make it clearer in the camera-ready version upon acceptance.
> > >
> > > Regarding classifier-free guidance (which we cite as a similar work), our derivation resembles theirs. However, we utilize it for a different purpose (yielding PAG), as you have stated.

---

### Official Review · Reviewer_gfgX · 2022-10-23

**Confidence:** 5
**Correctness:** 3
**Technical Novelty And Significance:** 3
**Empirical Novelty And Significance:** 4
**Recommendation:** 5

**Clarity, Quality, Novelty And Reproducibility:**

I am generally happy with the quality and novelty of this work. There are few concerns mentioned in the weaknesses part that I hope to be addressed during the discussion period.

**Strength And Weaknesses:**

Strengths:

- The problem that this paper is analyzing is both important and interesting.

- The study of the complementary path, from perceptually aligned gradients to adversarial robustness, is somewhat novel.

- Approaches considered in this paper for encouraging the gradients to be perceptually aligned are intuitive.

- Experimental results supports the claim of this work.

- The paper is well-written and easy to follow.

Weaknesses:

Despite that there are several aspects of this work that I like and appreciate, there are several concerns that need to be addressed.

1 - The main weakness of this work is the manual inspection of perceptually aligned gradients. Is there a way, based on the experimental results of this work, to quantify how aligned are the gradients of a given model? Conducting manual inspection might be misleading as it highly depends on the initialization.

2- Since all proposed approaches in this work do not include adversarial examples in their training routine, it is necessary to have a runtime comparisons between each proposed method and conducting adversarial training (it would also be fair to include other variants of adversarial training such as Free AT [A] and Fast AT [B]).

3- The experimental results is missing a large-scale real-world dataset such as ImageNet.

4- Analyzing the effect of encouraging the feature space to be semantically interpretable on adversarial robustness has been explored before in [C] through deploying metric learning. It would be nice to include describe the main differences between both this work and previous approaches.

5- While the proposed SBG matches the adversarial robustness of adversarially trained models, how would the performance be of we combine both approaches. For example, how would AT+SBG perform? The authors might want to explore more powerful variants of adversarial training such as AWP [D].

I generally admire the efforts in this work. I am also happy to increase my score based on the discussion with authors.

[A]: Adversarial Training for Free!, NeurIPS 2019

[B]: Fast is Better than Free: Revisiting Adversarial Training, ICLR 2020

[C]: Rethinking Clustering for Robustness, BMVC 2021

[D]: Adversarial Weight Perturbation Helps Robust Generalization, NeurIPS 2020

**Summary Of The Paper:**

This paper studies the effect of encouraging the semantic space of a deep model to be perceptually aligned on the adversarial robustness of the model.
This paper is based on earlier observations that robustly trained models have interpretable feature space.
This work encourages the gradients the model to be perceptually aligned through four different approaches.
Experimental results show that the obtained model, without training on adversarial examples, exhibits non-trivial robustness characteristics that could match the adversarially trained one.

**Summary Of The Review:**

This work has several merits that I appreciate. However, the main concerns regarding this work are:
(1) A systematic approach in quantifying the ''perceptuality'' of gradients of a given model.
(2) Large scale experiments to demonstrate the effectiveness of the proposed approach.
(3) Discussion on how related this work is with earlier works in the literature analyzing the complementary path (from semantics to robustness).

---

> ### Author Response · Authors · 2022-11-19
> **Authors' reply**
>
> We thank the author for the constructive feedback. We address below every raised point and revise our manuscript accordingly.
>
> 1) When conducting PAG assessments, we initialize the iterative deterministic process from the same base source image for all models, making this qualitative inspection stable and not misleading. Currently, there are no metrics for assessing PAG quantitatively. Generally speaking, when quantitatively assessing the performance of computer vision algorithms, we do so by comparing their results with a held-out test set containing ground-truth images and labels. However, when considering PAG, there are no "ground-truth" realizations of it, which makes a quantitative evaluation on a test set infeasible. However, we report the alignment level with the target gradients (i.e., SBG), but it can only serve as a proxy, as these are not ground-truth ones. We add to our paper a figure that shows qualitative performance alongside the robust performance using SBG (figure 5). Increasing the regularization coefficient ($\lambda$) leads to gradients more aligned with SBG. This visualization shows that better PAG with better alignment leads to more robust models.
>
> 2) Contrary to PGD iterations in AT that have to be sequential, our class iterations are parallelizable, thus enabling better scaling and faster computation. To illustrate this point, we measure the runtime of both our training algorithm and AT on a single GPU. We find that our method is x6.14 times faster than AT on CIFAR-10 which uses the standard PGD-20. We include this information in Appendix G of the revised paper.
>
> 3) Indeed, including a large-scale dataset such as ImageNet could better support our claims.  As we cannot employ our method to ImageNet in the rebuttal time window, we apply it to CIFAR-100 using both CLass-Mean and SBG and to Tiny ImageNet using SBG. We choose these as they have a large number of classes, thus consisting of more varied content. We include our results on these datasets in the revised version of the paper (Tables 3,5) and we are currently running more experiments (e.g., SBG for Tiny ImageNet.) We aim to include these and more (better optimization and more architectures besides ResNet-18) in the camera-ready version, upon acceptance. However, we believe that since we used multiple architectures types (please see also Appendix C) and four datasets, our findings are consistent. Moreover, many works (including SOTA ones) in the field abstain from utilizing ImageNet and focus on CIFAR datasets.
>
> 4) We thank the reviewer for noting this interesting work. We cite [C] and discuss its connection to ours in Appendix A.
>
> 5) We thank the reviewer for this proposal - we conducted an experiment that introduces our PAG promoting regularization to AT and discover that it can significantly enhance its robustness. This is surprising as adversarial robust models have PAG, nevertheless, improving it yields additional robustness. We report the results of this experiment in Section 5.4. We believe this positive discovery can be further utilized to improve robustification algorithms in the future.

---

### Official Review · Reviewer_gXjD · 2022-10-31

**Confidence:** 3
**Correctness:** 3
**Technical Novelty And Significance:** 2
**Empirical Novelty And Significance:** 2
**Recommendation:** 3

**Clarity, Quality, Novelty And Reproducibility:**

The writing is not very clear. The novelty is good but maybe it is because verifying the usefulness of PAG is not so attractive.

**Strength And Weaknesses:**

strength:
+ it is interesting to see several attempt to design several approaches for the ground truth gradients for alignment.

weakness:
- PAG provides additional supervised information and I do not think people have too much doubt about this. Meanwhile this could be explained directly rather than numerical experiments.
- In numerical experiments, the proposed methods do not show clear advantages over AT, expect for ViT model.  One question is about the performance of ViT, which seems quite weak, but in general ViT is believed to be more robust than CNN.
- The lack of ImageNet makes the conclusion less convincing. Besides, additional structures should be considered. In my opinions, to support the conclusion, typical NN structures (skip or not, dense or not, wide, different depth) and AT schemes should be included.

**Summary Of The Paper:**

In this paper, the authors throw out a question about the usefulness of Perceptually Aligned Gradients (PAG) and give a positive answer by numerical evaluation.

**Summary Of The Review:**

The problem this paper cares about is not very interesting. Even for this question, the answer could come from not only numerical experiments. Besides, the experiments are not sufficient. Overall, I tend to give negative score on this paper.

---

> ### Author Response · Authors · 2022-11-19
> **Authors' reply**
>
> We thank the author for the feedback. We address below every raised point and revise our manuscript accordingly.
>
> * We emphasize that PAG does not require any additional annotated data or labels. The main goal of the paper is to discover whether promoting PAG improves robustness. It was previously unknown that if the model's pointwise gradients point to a perceptually meaningful direction, it will be more robust (i.e., lack of misclassified samples in the surrounding of it). This finding uncovers underlying connections between PAG and robustness beyond simple additional supervision. Moreover, our work is significant and novel, and it is not the case that "people do not have too much doubt about this", as noted by other reviewers. For example, "The problem that this paper is analyzing is both important and interesting" [gfgX] and "Creating models with PAG which are not a byproduct of robustifying a model has to my knowledge not been done and is a significant contribution of the paper" [FuwE].
> We do not know how our research question can be explained directly, can you please elaborate on this? Such constructive feedback can help us better motivate our method before empirically demonstrating it with numerical experiments. In the paper, we provide the theoretical motivations and intuitions that led us to this research.
>
> * While AT uses adversarially perturbed images, we do not train on adversarial examples at all and still obtain comparable results. Methods for obtaining robust models without adversarial training are considered very interesting ("This brings forth the question – Could we use these insights to train robust classifiers with standard methods (i.e. without performing adversarial training)?" [1]). We provide such a method and discuss its characteristics and implications. Such research deepens our understanding of adversarial robustness, and paves the way for a more careful assessment of it, in addition to simply comparing the robustness accuracy performance of different models. Moreover, we perform better on some AT failure cases -- small datasets (see STL) and ViT. ViT is indeed believed to be more naturally robust, and our "vanilla" column shows it. However, our empirical study demonstrates that AT is better suited for CNNs and is not optimal for ViTs. Thus, using an off-the-shelf AT algorithm to ViT performs worse than our approach.
>
> * In the experimental part of the paper, we thoroughly analyzed different architecture types and datasets using all of the proposed methods. Thanks to your request we’ve added experiments to show the method on more diverse datasets, CIFAR-100 and Tiny ImageNet. These initial results are promising and we aim to run additional experiments for obtaining even better results for the camera-ready. Moreover, we have analyzed the effects of our approach using additional architectures and append the results in Appendix C. Our new results correspond with the main findings of the paper, and we believe that they verify the discovered connection between PAG and robustness in neural networks. We thank the reviewer for this suggestion and hope that these findings address their concerns.
>
>
>
> [1] Robustness may be at odds with accuracy (https://arxiv.org/abs/1805.12152)

---

> > ### Comment · Reviewer_gXjD · 2022-11-25
> > **thank for the reply**
> >
> > Thanks for the reply and discussions.
> >
> > Indeed, aligning the gradients to other gradients, including the OI, CM, NN, and SBG, is different to the previous methods that aligns gradients of clean and perpetuated examples. BUT, the essence is the same: a regularization term on gradient from other images. In this point of view, using adversarial examples is even more meaningful, since nature examples and its  adversarial ones, not others, have more perceptual similarity.
> >
> > As a result, it is good that the proposed method does not use adversarial examples, however, the final performance is the same: nature accuracy is scarified for better robustness. In other words, it seems that adversarial examples are avoided, but the same bad things happen, which again supports my previous thinking: the essence is the same. From the reported result, the trade-off performance is even more significant: in Table 1, the proposed method sometimes has good robustness but the clean accuracy is clearly worse than AT. In Table 3 and Table 7, similar results could be observed that one with lower clean accuracy has better robustness.
> >
> > Previously, I have questions on the experiments. Thanks for some additional results and discussions. However, I still have the following concerns. (1) there is no significant advantages over AT (avoiding using adversarial examples is indeed good, but the aim is to avoid drop on clean accuracy. If there is still drop, this advantage becomes less interesting). (2) the performance of ViT is still puzzling. Even for Vanilla training on CIFAR-10, ViT has only 80.51% accuracy, too low in my opinion.

---

> > > ### Author Response · Authors · 2022-11-25
> > > **Authors reply**
> > >
> > > Thank you for your reply.
> > >
> > > * **Difference from Adversarial Training (AT):** There are several essential differences between our proposed methods and AT. The latter minimizes the cross-entropy loss on worst-case perturbed examples in an $\epsilon$-ball without a regularization term. These instances are obtained via iterative gradient descent. AT procedure leads to small input gradients in the volume around the data points, regardless of the direction of the gradient.
> > > Crucially, our framework is substantially different: We do not utilize iterative gradient descent, **nor do we train the model on examples outside the training set**. Instead, we penalize a point-wise property using a regularization term **on the training data points only**, and we do not consider *"gradients from other images"*. Specifically, we **penalize over the direction of the gradient and not its magnitude**. This makes our findings more surprising, as we show significant $\epsilon$-ball robustness across various architectures and datasets without training on adversarially-perturbed examples.
> > >
> > > * **Clean accuracy – robustness tradeoff:** It is widely believed that robustness is intrinsically at odds with clean accuracy [1, 2]. Thus, our method suffers from this tradeoff as well. It does not mean that the essence of the two methods is the same, but rather that the resulting robustness effect is similar: they both lead to robust but less accurate models.
> > > In Figure 5 of our revised paper, we show an interesting and meaningful way to trade off clean accuracy for robustness by changing the levels of gradient alignment via $\lambda$. For example, $\lambda=0.1$ leads to $22.53$ $l_2$ robustness with $91.32$ clean one, compared to $0$ and $93.61$ of the vanilla classifier. When seeking more robustness, one can increase $\lambda$, e.g., $\lambda=2$ leads to $55.39$ and $78.56$. We will also report the clean accuracies specified here and further elaborate on this discussion in the camera-ready version upon acceptance.
> > >
> > > * **Advantages over AT:** We would like to reiterate that the goal of this paper is not to provide an alternative practical solution for robustness but rather to examine the connection between PAG and robustness – an interesting question that was yet to be studied and is fascinating to the research community [1]. Nevertheless, our method has several advantages over AT: it is more computationally efficient and performs better under low-data regimes.
> > >
> > > * **ViT performance:** ViTs usually benefit from large amounts of data pretraining, which we do not have in CIFAR-10. Moreover, the division to patches in a low-resolution dataset as CIFAR-10 is less effective than in higher-resolution ones. We utilized an off-the-shelf ViT to compare to the baselines using the same setup, guaranteeing the fairness of the comparison. Finally, we provide our source code, enabling the reproduction of the reported results. In addition, to further address your concern, we are currently running more ViT experiments on STL, CIFAR-100, and Tiny ImageNet and will report their results in the camera-ready version.
> > >
> > > [1] Robustness may be at odds with accuracy (https://arxiv.org/abs/1805.12152)
> > >
> > > [2] Is Robustness the Cost of Accuracy? (https://arxiv.org/abs/1808.01688)

---

### Official Review · Reviewer_FuwE · 2022-11-01

**Confidence:** 4
**Correctness:** 3
**Technical Novelty And Significance:** 3
**Empirical Novelty And Significance:** 3
**Recommendation:** 5

**Clarity, Quality, Novelty And Reproducibility:**

The paper is extremely clear and easy to understand. The main idea being conveyed is simple and problem under study is novel.

**Strength And Weaknesses:**

Strengths:
1. Creating models with PAG which are not a byproduct of robustifying a model has to my knowledge not been done and is a significant contribution of the paper.
2. The main idea of the paper is extremely simple and the presentation is straightforward. The paper on the whole is well written and clear.
3. Experiments in the paper support the authors claim that PAG can imply robustness (see concerns below).
4. Results in the L2 attack case are very strong. The authors observe robustness better than AT in some cases without the additional overhead of robust training.

Weaknesses:
1. My main concern about the results in the paper center around the lack of quantification of PAGs. There is a huge difference in robustness against L2 and L-infinity attacks (shown in Table 1). It seems there is little robustness induced (especially in the heuristic L-infinity VIT case) in some settings even when there are PAGs (as shown in Fig 4). Therefore, it seems that even though PAG is present at least superficially, no corresponding robustness is observed. This goes against what the paper is trying to prove (that PAGs imply robustness). I’m not sure if this only shows up in one case (heuristic L-infinity VIT case) but there is some explanation needed.
2. Related to the above point, since there is no quantitative measure of PAG, it is hard to understand whether there is a dose-response relationship between PAG and robustness. If there were a dose-response relationship, it could potentially explain my concern above (Pt 1) by saying that - the PAG for the heuristic L-infinity VIT case is weak so robustness is not very high. The authors already use cosine similarity in the loss which could potentially serve as a quantitative metric for the PAGs.
3. I feel the paper is lacking analysis to definitively “prove” that PAGs imply robustness. For example, we have seen improving PAG improves robustness, but what happens if we purposely try to decrease PAG? (flip the sign on the regularization term). What happens if train a model using AT and add a regularization term to decrease PAG? (i.e. essentially increase robustness and decrease PAG simultaneously). Without additional analysis, I think the paper falls short in showing where and how the effect being claimed exists.

**Summary Of The Paper:**

This paper aims to create image classification models which produce gradients which are “perceptually aligned” (in the sense of Tsipras et al.). The authors then essentially try to understand whether this alignment can imply robustness (i.e. the opposite of the claim first made in Tsipras et al.). The main difficulty is to produce a model with PAG (perceptually aligned gradients) without inheriting robust characteristics as a byproduct from the adversarial training (AT) process.

The authors propose a technique to induce PAG by adding a regularization term to the usual cross entropy loss of an image classification model. This term essentially forces the gradient of the model to point in the direction of the “ground truth” PAG via cosine similarity.

There are two ways in which these ground truth PAGs are obtained:
1. Heuristic: The model is forced to point its gradients towards a general representative (r_t) of the target class (gradient = r_t - x). The representative can be chosen three ways: pick an arbitrary image from the class (OI), choose class mean (CM) or take the nearest neighbor image to x (NN).
2. Score based: A pretrained diffusion model is used to obtain an estimation for “ground-truth” classifier input-gradients by a simple subtraction of conditional and unconditional outputs of the network .

PAG is evaluated qualitatively via images of one class morphing to another class (See Fig 4). Experiments are performed on a toy 2d dataset, CIFAR-10 and STL with a ResNet model and a ViT. Results show strong robustness results (at least against L2 attacks) for all models/datasets.

**Summary Of The Review:**

Overall, I think the paper makes a solid attempt at understanding a curious property of image classification models. I think the approach introduced by the authors is very interesting. However, there is a lack of analysis discussing the limitations of the technique being proposed which keeps me from recommending publication.

---

> ### Author Response · Authors · 2022-11-19
> **Authors' reply**
>
> We thank the author for the constructive feedback. We address below every raised point and revise our manuscript accordingly.
>
> * (1-2) Unfortunately, there are currently no methods to quantify PAG. While cosine similarity indeed measures the distance between our target and the model's gradient, the former ones are not ground truths, making it just a proxy to PAG. As stated correctly, the results in Table 1 show that our method leads to lower robustness in $l_infty$ compared to $l_2$. As we state in the paper, we hypothesize that the performance difference between the two stems from the cosine similarity. This trend is general and not specific to ViT. The results in Table 1 and Figure 4 show different levels of PAG and different levels of robustness. It suggests that PAG is not a binary trait, and there are levels of it. For example, in the heuristic $l_\infty$ ViT case, the PAG is very limited and so is the robustness performance. To better demonstrate that a positive correlation exists (i.e., better PAG implies more robustness), we conduct an experiment where we use SBG and different values of $\lambda$. Our findings show that higher values of $\lambda$ lead to better PAG, as expected, but also to much more robust models. We've updated the manuscript accordingly (Figure 5).
>
> * (3) These are important and interesting notes, but although we initially viewed them as deviating from our research question, we conduct such experiments. Regarding flipping the sign of our regularization term, it is not guaranteed to reduce PAG, as vanilla models do not possess PAG. Thus, by comparing vanilla ones with classifiers trained using our objective, we verify that, indeed, PAG leads to improved robust performance. Moreover, thanks to your comment, we've conducted an experiment, revealing that increasing PAG increases robustness, further strengthening our discovery. In addition, while incorporating PAG regularization into AT also deviates from our main question, we conduct experiments examining the effect of introducing our PAG-inducing regularizer into AT, to examine the potential of our approach to enhance robust training. Interestingly, we find out that although AT yields to PAG, plugging our regularization leads to better PAG and to more robust models (section 5.4). We believe that these experiments better verify the connection between PAG and robustness and we thank the reviewer.

---

### Official Review · Reviewer_XeGZ · 2022-11-02

**Confidence:** 4
**Correctness:** 3
**Technical Novelty And Significance:** 2
**Empirical Novelty And Significance:** 3
**Recommendation:** 6

**Clarity, Quality, Novelty And Reproducibility:**

The paper is easy to read and follow. In general, the paper is original and timely as it tackles an important problem. The proposed approach has some interesting properties compared to adversarial training, such as good performance in the low-data regime and a lower complexity of training compared to adversarial training. However, some of the authors’ claims are unsupported. For example, the authors claim that the proposed method provides robustness on par with adversarial training. However, the comparison does not include more recent methods which address some of the issues of adversarial training.

**Strength And Weaknesses:**

### Strengths

- Poses and attempts to answer the question: do perceptual gradients imply adversarial robustness?
- Novel model training called Perceptually Aligned Gradients that attains adversarial robustness without running an expensive adversarial attack, such as PGD-10, in the inner loop of the model training.

### Weaknesses

- The objective for optimizing alignment requires computing cosine similarity between all classes, which scales poorly for problems with many classes. The authors didn’t present results for problems with more than 10 classes.
- The lack of a theoretical analysis of the proposed method. It is not clear why the proposed objective will guide the training toward a more robust model.
- The comparison was made only with vanilla Madry adversarial training. The authors should include a comparison with more recent adversarial training methods, e.g. TRADES, FAT, Early Stopping PGD, etc.

### Questions

- How does the proposed method compare to Lipschitz regularization methods which require a small input gradient? Which approach is more suitable for robust training?
- Could the authors include experimental comparison with denoising methods using diffusion models? Why the presented approach is more suitable for robust training?

**Summary Of The Paper:**

The authors pose and attempt to answer a new question if the perceptual gradients of the model imply the robustness of the model to adversarial attacks. To answer that, they proposed a new method that trains the model to have input gradients that align with ground-truth input gradients. They introduced several methods to obtain ground truth gradients. In the experiments, they showed that using perceptual input gradients as a constraint alone is sufficient to achieve robustness comparable to adversarial training.

**Summary Of The Review:**

The authors attempt to answer the question if perceptual gradients imply adversarial robustness. They proposed a novel training method that directly optimized the perceptual gradients. Using the proposed method, they showed that perceptual gradients indeed imply robustness.  The idea is novel and interesting. However, some of the claims might not be well supported and the authors should expand both theoretical and empirical analysis of the proposed approach.

---

> ### Author Response · Authors · 2022-11-19
> **Authors' reply**
>
> We thank the author for the constructive feedback. We address below every raised point and revise our manuscript accordingly.
>
> **Weaknesses:**
>
> * As you have correctly stated, we present an optimization problem that requires iterating over all classes. However, unlike PGD iterations in AT that ought to be sequential, our class iterations are parallelizable, thus enabling better scaling. To illustrate this point, we measure the runtime of both our training algorithm and AT on a single GPU. We find that our method is x6.14 times faster than AT on CIFAR-10 which uses the standard PGD-20. As for the issue of scalability, we apply our method using SBG and CM to CIFAR-100 and Tiny ImageNet using CM, which contains 100 and 200 classes, respectively. Our method performs well on these datasets as well, achieving 29.25% and 19.21% in $l_2$ robustness, respectively. We've updated the manuscript to include such experiments. However, we believe that these results can be further improved with more experiments that we are currently running and we will include them in the camera-ready version.
>
> * Indeed, it is unintuitive that enforcing a model to have pointwise gradients aligned to some target should yield robust models. To provide some intuition on the connection between PAG and robustness, we conduct experiments on a toy example. Moreover, we hypothesize that our method encourages the model to prefer robust features over non-robust ones, as termed in [1].  Since our work is the first to uncover robustness features by simply encouraging PAG, we defer more rigorous theoretical proofs to future work.
>
> * We thank the reviewer for his suggestion and we will do so in the camera-ready version. Nevertheless, we do not claim to outperform SOTA robustification methods. We show that our adversarial-free method yields surprising robustness, even comparable to the basic AT.
>
> **Questions:**
>
> * Indeed, at a first glance, there is a resemblance between Lipschitz regularization methods and our approach, as both penalize over the input-gradients. However, there are a few major differences – first, while such methods penalize the norm of the gradients and not their direction, we do the complete opposite. Second, these methods are known to yield adversarial robustness and are connected with adversarial training, thus cannot be used to answer our research questions. Lastly, these methods are not shown to lead to PAG. We include such a discussion in Appendix A of the revised paper.
>
> * While diffusion models have been utilized for adversarial purification, as a preprocessing step before feeding the images into the classifier, we take a completely different approach – we use the amazing capabilities of such models to estimate the score function and distill target PAG gradients for our objective. Amazingly, we prove that forcing models to be aligned to such gradients leads to substantial robustness.
>
>
>
>
>
>
>
> [1] Robustness may be at odds with accuracy (https://arxiv.org/abs/1805.12152)

---

### Author Response · Authors · 2022-11-19
**Authors' general notes**

We sincerely thank the reviewers for their constructive feedback and notes. During the rebuttal period, we conducted extensive experiments based on your suggestions and updated the manuscript accordingly. The new results address most of the raised issues and further improve the paper. The main modifications are as follows:

* We added experiments on CIFAR-100 (100 classes) and Tiny ImageNet (200 classes) for demonstrating the method’s scalability and to validate the connection between PAG and robustness in more diverse real-world datasets. We report CIFAR-100 and Tiny ImageNet results in Tables 3 and 5 in the revised paper, respectively. We note that these are initial results that can be further improved with more extensive experiments, which we will provide for the camera-ready version of the paper (e.g., SBG on Tiny ImageNet and further optimizations).
* We added experiments that empirically prove the positive correlation between PAG and robustness: more aligned gradients induce more robust models (Figure 5).
* Enhancing Adversarial Training via PAG: introducing our novel objective to AT yields significant improvement for both $l_2$ and $l_\infty$ threat models (Section 5.4 and Table 6). We aim to provide a more thorough study using more advanced robustification methods for the camera-ready.
* We added experiments on more architectures – VGG a CNN without skip connections and MLP-Mixer, a dense one, and report the results in Table 4 in Appendix C. These results empirically validate that the connection between PAG and robustness is architecture agnostic. We aim to conduct more such experiments and add them to the main paper of the camera-ready version.

Moreover, we wish to emphasize that the main goal of this paper is to disentangle PAG from adversarial training and study the connection between the two.
Thanks to the reviewers' feedback, we conduct additional studies that further verify this connection. While our approach leads to significant robustness comparable to AT, we defer the improvement of non-adversarial robustification methods for future work.

---

### Decision · Program_Chairs · 2023-01-20

**Decision:**

Reject

**Justification For Why Not Higher Score:**

Main weaknesses of this paper include:
(1)Even PAG is combined with AT, the robust accuracy is significantly lower than SOTA (https://robustbench.github.io/#div_cifar10_L2_heading).
(2)The proposed method did not show a promising Clean accuracy – robustness tradeoff.

**Justification For Why Not Lower Score:**

N/A

**Metareview: Summary, Strengths And Weaknesses:**

Motivated by the observation that some earlier works have identified Perceptually Aligned Gradients (PAG) as a byproduct of robust training, but none have considered it as a standalone phenomenon nor studied its own implications, the authors studied if the perceptual gradients of the model imply the robustness of the model to adversarial attacks.
They proposed to directly promote PAG in training classifiers and examine whether models with such gradients are more robust to adversarial attacks. They showed that using perceptual input gradients as a constraint alone is sufficient to achieve robustness comparable to adversarial training.
Overall, the authors have properly responded to reviewers' comments (although some reviewers did not actively participate in the rebuttal process).

Main weaknesses of this paper include:
(1)Even PAG is combined with AT, the robust accuracy is significantly lower than SOTA (https://robustbench.github.io/#div_cifar10_L2_heading).
(2)The proposed method did not show a promising Clean accuracy – robustness tradeoff.

Based on the above concerns, this paper is recommended to be rejected.